# Attosecond optoelectronic field measurement in solids

Shawn Sederberg[1], Dmitry Zimin[1,2], Sabine Keiber[1,2], Florian Siegrist[1,2], Michael S. Wismer[1,2], Vladislav S. Yakovlev [1,2], Isabella Floss[3], Christoph Lemell [3], Joachim Burgdörfer[3], Martin Schultze [1], Ferenc Krausz[1,2] & Nicholas Karpowicz[1]*

The sub-cycle interaction of light and matter is one of the key frontiers of inquiry made accessible by attosecond science. Here, we show that when light excites a pair of charge carriers inside of a solid, the transition probability is strongly localized to instants slightly after the extrema of the electric field. The extreme temporal localization is utilized in a simple electronic circuit to record the waveforms of infrared to ultraviolet light fields. This form of petahertz-bandwidth field metrology gives access to both the modulated transition probability and its temporal offset from the laser field, providing sub-fs temporal precision in reconstructing the sub-cycle electronic response of a solid state structure.

[1] Max-Planck-Institut für Quantenoptik, Hans-Kopfermann-Straße 1, 85748 Garching, Germany. [2] Department für Physik, Ludwig-Maximilians-Universität, Am Coulombwall 1, 85748 Garching, Germany. [3] Institute for Theoretical Physics, Vienna University of Technology, Wiedner Hauptstraße 8-10, 1040 Vienna, Austria. *email: nicholas.karpowicz@nanotec.cnr.it

The use of light fields to control electronic devices has fundamentally transformed communication and computing. The production of pairs of charge carriers through the absorption of light in a semiconductor is a fundamental interaction linking light and electronics. Modulation of the light illuminating a photosensitive structure in turn modulates the electronic transition rate, transferring information from the light field to the optoelectronic device. One may ask then, is there any intrinsic latency associated with photoelectric and photovoltaic processes on the timescale of the optical field and, likewise, might this impose fundamental speed limits for information processing? That is, under which conditions does the assumption that the rate of charge-carrier injection responds instantaneously to changes in the optical intensity break down? It has been anticipated before that real-time, sub-femtosecond resolution of the coupling between light and electrons inside macroscopic bulk solids might be achievable through the use of light-field-induced currents resulting from rapid changes to the electronic properties of solids induced by strong laser fields[1]. We demonstrate in this communication that these changes can be employed as a gate to record the field evolution of electromagnetic waveforms with extremely high temporal resolution, providing petahertz-bandwidth field metrology. Comparison of experimental data with simulation results allows us to access the underlying electronic dynamics through their influence on the ultrafast gate function.

The use of controlled laser fields[2] has led to many ground-breaking observations in the interaction of light with solids[3–18]. Much of this work employed waves in the terahertz and mid-infrared spectral ranges, where techniques such as electro-optic sampling[19,20] (EOS) and photoconductive switches[21] allow straightforward access to the time-varying electric fields employed. While EOS[22,23] and coherent control[24] hold the potential for pushing the frontiers of field sampling in solids, experiments studying light fields in the visible to ultraviolet range[3,11,12] and providing sub-femtosecond resolution have so far required attosecond time gating provided by the high-harmonic generation process[25–29] or tunnel ionization[30] in gases. Attosecond metrology and spectroscopy have thus far been confined to systems in the gas phase[31,32], to material interfaces[33], and to nanometer-thin layers[3] of solids. Measurements within solids, the platform of modern electronics, remained beyond practical reach.

Being able to resolve femtosecond-scale field oscillations has recently allowed the observation of energy exchange between a strong laser field and a dielectric, which includes a nearly lossless transient flow of energy into and out of the material in the formation of the non-resonant nonlinear polarization, and an irreversible flow of energy into the material as charge-carrier pairs are generated through nonlinear absorption at higher intensities[34]. Simulations and experiments show that this energy exchange does not smoothly follow the time-averaged envelope of the laser pulse; it is distinctly modulated due to the sub-cycle dynamics of the light–matter coupling. We aim at exploiting this staccato flow of energy to control electronic devices with light fields on a sub-femtosecond timescale.

Here, we demonstrate the precise temporal characterization of the electronic response in a wide energy gap material on the attosecond scale in macroscopic volumes of condensed matter, exploiting the fact that currents inside dielectrics may be switched in ultrashort time intervals during their interaction with strong fields[1,14,35,36]. The temporal evolution of the electronic response to a driving electric field could not be observed in these previous measurements due to the lack of a temporal reference. A direct link between the measured current and the driving electric field waveform could not be established or confirmed. We now show

by manipulation of the optical waveform that the highly non-linear interaction can be largely confined to a temporal interval shorter than 1 fs. The resultant attosecond temporal gate, along with controlled optical fields, offers all-solid-state technology for sensitive sampling of time-dependent electric fields with a >1 PHz detection bandwidth spanning the mid-infrared to ultraviolet. The measurement of the attosecond timing of the laser-induced currents in a solid state material yields a time-domain view on the absorption of strong light fields by dielectrics opening the pathway to unraveling the underlying excitation mechanisms. Most importantly, the electric field measurement is compared to a known electric field waveform, both establishing its accuracy and providing a direct link between the measured signal and the electric field under scrutiny. The resultant all-solid-state attosecond metrology provides real-time insight into light–matter interactions without the need for large vacuum beamlines and dedicated infrastructure but instead relying on compact and flexible tabletop setups.

## Results

**Optical field sampling in solids.** Exploiting a highly nonlinear process in a wide energy gap solid with a light pulse comprising a single oscillation cycle ($T_{FWHM} = 2.7$ fs, $\lambda_L = 750$ nm), focused to a field strength between 1 and 2 V Å$^{-1}$, well below the ~2.7 V Å$^{-1}$ damage threshold[3] of the material we aim at confining carrier injection to a single sub-fs interval. In wide band-gap materials, the highly nonlinear transitions effectively suppress carrier injection at wave crests other than one or two central half cycles of a few-cycle pulse, resembling the temporal confinement seen in strong field ionization of atomic systems. This temporal confinement relaxes the requirements on the duration of the injection pulse (see Supplementary Note 5 for detailed analysis) and increases the sampling resolution, enabling electric-field sampling up to the petahertz range with a simple solid-state circuit.

As depicted in Fig. 1, a broadband light pulse used for carrier injection, $E_i$, is incident on a material (crystalline quartz in this case) at sufficient intensity to induce transitions from the (filled) valence to the (empty) conduction band via high-order nonlinear optical processes, which predominantly occur near the pulse's highest field maximum. The injection field is s-polarized (perpendicular to the observed current) and hence accelerates charge carriers in the direction parallel to a pair of electrodes placed adjacent to the interaction region. We have confirmed that, on its own, the injection pulse does not contribute to the measured signal. A second light field, $E_d$, polarized orthogonally to $E_i$ (p-polarization), is applied collinearly and overlaps in time with the injection pulse, with a variable delay. This field, carried at a photon energy far below the band gap, is too weak ($E_d \ll E_i$; $E_d$ well below 1 V Å$^{-1}$) to induce nonlinear transitions[1], i.e., measurable macroscopic currents, on its own and negligibly affects the carrier injection process. Rather, $E_d(t)$ drives the carriers by primarily changing the horizontal component of the crystal momentum of the injected carriers.

Electrons and holes that are injected and then displaced in the band structure by the two pulses form a macroscopic dipole that can have sufficient amplitude and persist for a time long enough to induce a screening signal and thus a measurable current in the pair of electrodes[37]. Altering the delay $\tau$ of the driving pulse relative to that of the injection pulse modifies the induced dipole, allowing for the readout of the residual screening signal, $S_d(\tau)$, by means of standard radio-frequency electronics. As we show below, this signal is directly related to the electric field that caused the charge separation, $E_d(t)$, which we measure with attosecond temporal resolution.

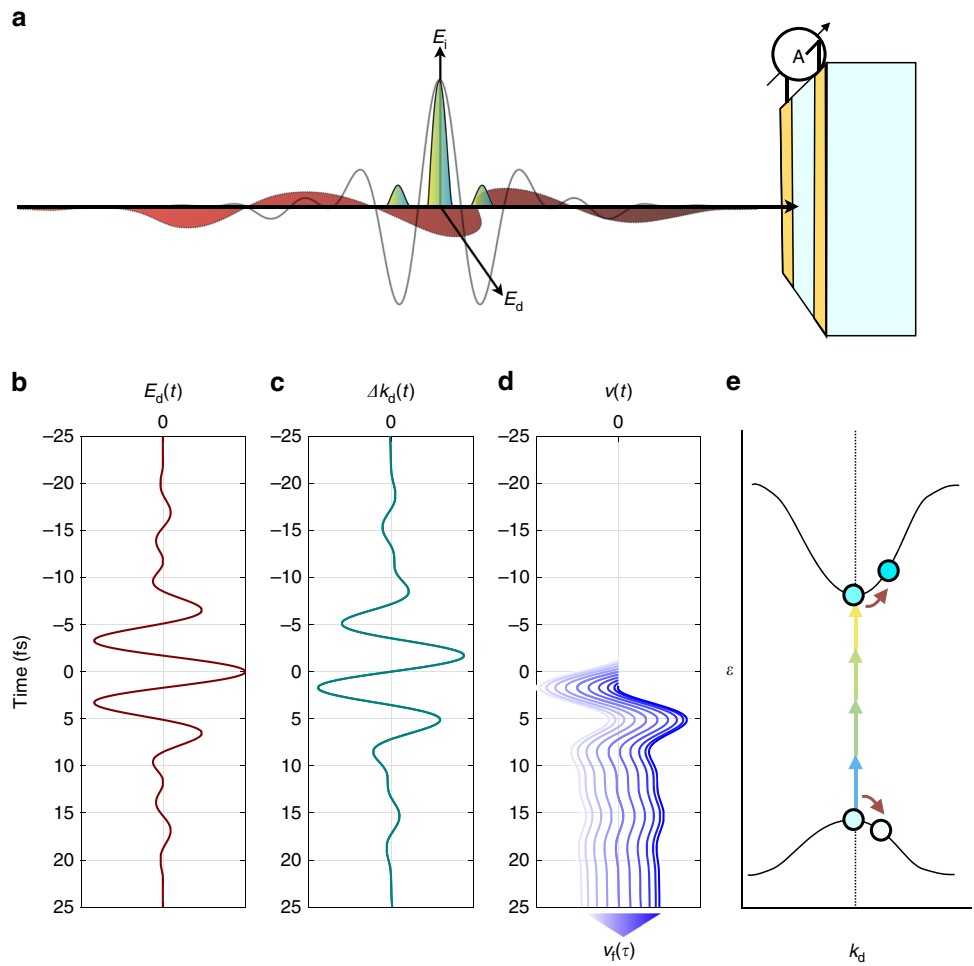

**Fig. 1 Field-dependent motion of charge carriers upon interaction with a pair of laser pulses. a** The cross-polarized fields, $E_i$ and $E_d$ are incident on a Z-cut quartz sample with gold electrodes attached to the material in the vicinity of the point of incidence of the laser pulses. The strong, nearly single-cycle pulse $E_i$ induces a transition from the valence to the conduction band in the material, in the presence of the orthogonally polarized driving field $E_d$ (**b**). **c** In an independent-particle picture, the crystal-momentum offset $\Delta k_d$ of an injected carrier along the horizontal direction is proportional to the time integral of the field $E_d$. This offset, together with band energies, determines the group velocities of charge carriers (**d**). In a semiclassical picture where photoinjection at time $\tau$ creates a wave packet with zero initial velocity, the average velocity at later times is determined by $\Delta k_d(t) - \Delta k_d(\tau)$. Thus, the average velocity of the carriers is sensitive to the time $\tau$ of the transition, which is controlled experimentally via the relative delay of the two laser fields, setting the in-band acceleration of the carriers after the photoinjection event (**e**). The measured dipole in the dielectric is proportional to the average displacement of charge carriers, which is obtained by integrating their average velocity over time.

As long as the driving field is so weak that the response to it is linear, the measured signal $S_d(\tau)$ can be written as a convolution of the vector potential

$$A_d(t) = \int_t^{\infty} E_d(t')\, \mathrm{d}t' \tag{1}$$

of the driving pulse (Coulomb gauge; displacement in band structure, i.e., crystal momentum) with a gating function, $G(t)$, which represents the excitation dynamics caused by the injection pulse, as well as physical processes leading to the formation and detection of the macroscopic dipole:

$$S_d(\tau) = -\int_{-\infty}^{\infty} A_d(t)G(t-\tau)\, \mathrm{d}t . \tag{2}$$

When $G(t-\tau)$ is confined to a temporal window much shorter than one half-cycle of the field, $\mathrm{d}S_d/\mathrm{d}\tau$ will take the form of $-A'_d(t) = E_d(t)$. Any device with signal characteristics given by Eq. (2) is capable of recording time-dependent electric fields with a bandwidth dictated by the inverse of the duration of carrier

injection. We refer to this concept as nonlinear photoconductive sampling (NPS). Its implementation with near-single-cycle near-infrared light is sketched in Fig. 1. The carrier injection timescale is determined by the injection pulse duration, and places an upper limit on the applicable pulse duration for broadband operation, as discussed in Supplementary Note 5, leading to a limit of $\approx 3.8$ fs at a wavelength of 800 nm.

Previous experiments[1,35,36] demonstrating sub-cycle control of electric currents showed that the interaction was triggered on a sub-cycle timescale and that the carrier-envelope phase of ultrashort laser pulses can be measured. Here we provide experimental evidence that NPS with a sufficiently short injection pulse enables optoelectronic measurements of electric fields oscillating at infrared, optical, and ultraviolet frequencies. We test and verify the relationship between the measured signal, $S_d(\tau)$, and the driving field, $E_d(t)$, by comparing results of NPS measurements to those of attosecond streaking[38] and electro-optical samling[19,20]—well established methods of direct measurements of field oscillations.

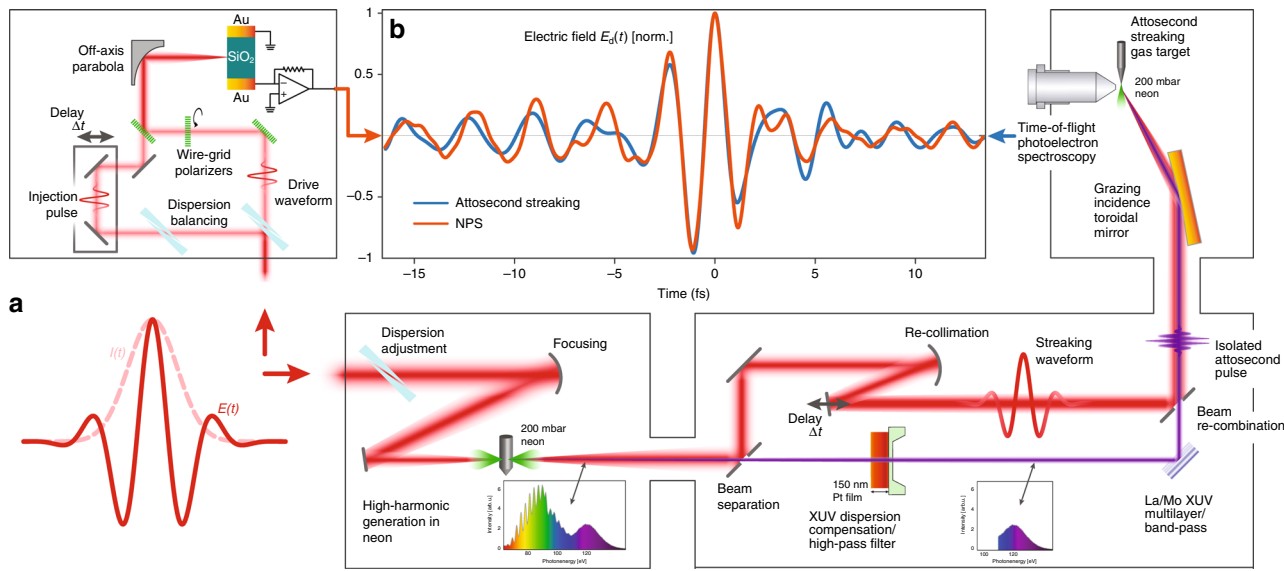

**Fig. 2 Field metrology via attosecond streaking and nonlinear photoconductive sampling (NPS).** The same few-cycle laser pulse (**a**) is sent to either the NPS setup or to an attosecond beamline. The NPS setup produces time-delayed replicas of the pulse with crossed polarizations, focused onto the quartz sample with adjustable time delay, while the signal induced at the end of their interaction with the solid in an external pair of electrodes is read electronically. In the attosecond beamline, attosecond streaking is used to obtain a comparative measurement of the temporal evolution of the laser field, as it provides a broadband response in the relevant frequency range. In attosecond streaking, the laser pulse to be measured is coincident on a medium where bound electrons can be ejected by a synchronized attosecond pulse of extreme ultraviolet (XUV) light. The shift of the photoelectron energy vs. time delay allows the direct measurement of the vector potential of the laser pulse. The waveform obtained via the time-derivative of the energy shift of the photoelectrons emitted from a neon atom in an attosecond streaking experiment is also retrieved via NPS, returning the driving field $E_d$. The results are remarkably similar given the differences between the two measurement setups (**b**).

We performed NPS and attosecond streaking measurements with the same light source. This source provided both the injection ($E_i$) and driving ($E_d$) fields for the laser-controlled electronic circuit. Despite significant differences in the underlying physics of each detection scheme (see brief description of attosecond streaking in the caption of Fig. 2) and the different optical beam paths, the electric fields returned by NPS and attosecond streaking exhibit significant similarity as shown in Fig. 2b, with a correlation coefficient of $\rho = 0.88$, and rms field error[39] $\epsilon = 0.48$. The two measurement systems include different focusing and beam transport optics, to which we believe a significant amount of the differences between the fields may be attributed.

The compact measurement system used for NPS, of a level of complexity comparable to systems which return the autocorrelation of the pulse, is sufficient to provide the injection and drive fields to the solid state sample, whose final state after the two-pulse interaction is read out using a lock-in amplifier. In contrast, performing a comparative measurement with attosecond streaking requires the use of an attosecond beamline, including attosecond pulse generation and isolation, and photoelectron spectroscopy using the resultant extreme ultraviolet photons.

The agreement between the waveforms retrieved using the two different methods confirms that nonlinear interband excitation effectively time-gates the carrier injection that contributes to the final charge offset. Moreover, it indicates that the injection process is substantially localized to a sub-cycle timescale. With the help of ab initio simulations we achieve a more detailed understanding of the timing of the carrier injection process.

**Sub-femtosecond carrier injection.** The frequency response of the nonlinear photoconductive sampler is intimately tied to the temporal evolution of the carrier injection process. Time-dependent density functional theory calculations[40,41] (TDDFT)

suggest the energy transfer rate being approximately proportional to the carrier injection rate, once the transient energy exchange responsible for the linear (refractive index) and third-order (Kerr effect) optical responses are removed (see Supplementary Note 4 for details). Performing such simulations for our experimental conditions and fitting the time-dependent excitation density without transient contributions to a function of the form $A \cdot E_i^{2n}(t - \tau_{inj})$ reveals a rate of energy deposition approximately proportional to the eighth power of the electric field ($n = 4$). Due to this highly nonlinear dependence, carrier injection is confined to brief, sub-half-cycle intervals synchronized to the extrema of the oscillating injection field, $E_i$. Thus, only one or two crests of the field contribute a significant number of charge carriers, depending on the carrier-envelope phase $\phi_{CE}$ of the injection pulse. This phase defines the temporal relationship between the envelope of a laser pulse and the underlying carrier wave, where

$$E_i(t) = \text{Re}[\tilde{F}_i(t) \exp(-i\omega_L t + i\phi_{CE})], \qquad (3)$$

$\tilde{F}_i(t)$ is the complex envelope of the injection field and $\omega_L$ is its carrier frequency. Modifying $\phi_{CE}$ changes the relative height of the two strongest wave crests. $\phi_{CE} = 0$ is defined as the case when the maximum of the carrier wave coincides with the maximum of the envelope function $|\tilde{F}_i(t)|$. For such injection pulses, carrier generation in TDDFT simulations is effectively confined to a single burst of sub-500-attosecond duration (full-width at half-maximum), see Fig. 3d.

As the TDDFT simulations do not provide a direct link between the microscopically induced dipole and the experimental screening signal $S_d(\tau)$ in the current, we propose a simple ansatz for the gating function from Eq. (2): guided by the TDDFT results, we approximate the gate by $G(t) \propto E_i^8(t - \tau_{inj})$, relating it to a nearly instantaneous rate of energy deposition but accounting for a small temporal displacement $\tau_{inj}$.

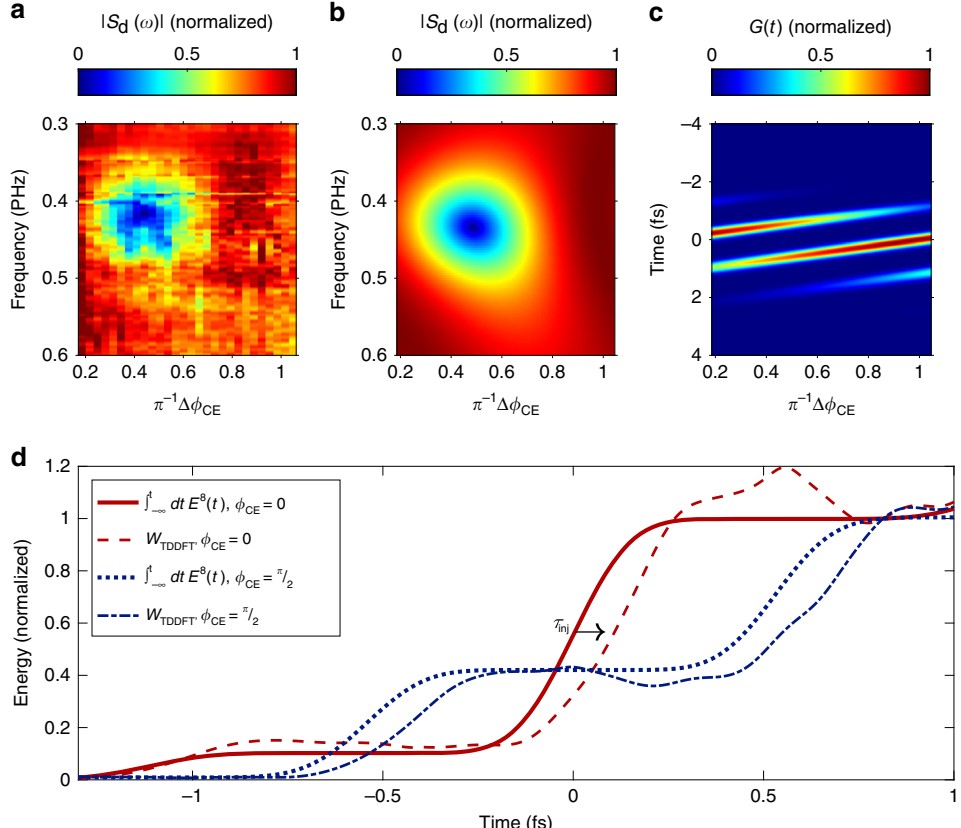

**Fig. 3 Sub-cycle modulation of transition rate observed via phase-dependent spectral response.** NPS waveforms were recorded over a series of values of $\phi_{CE}$ of the injection pulse and converted to the frequency domain by taking the modulus of its Fourier transform. Normalizing each spectral component removes the influence of the spectrum of $E_d(t)$ and shows the modulation depth of each frequency component. This normalized spectral response vs. phase **a** shows a strong, phase-dependent minimum at the 0.42 PHz central frequency of the injection pulse. This is consistent with a model simulation that uses $E^8(t)$ as the gating function (**b**, **c**). In this model, there is a single dominant photoinjection event for $\phi_{CE} = 0$, which results in a flat, unmodulated spectral response. For $\phi_{CE} = \pi/2$, there are two transition events of comparable magnitude spaced by one half-cycle of the laser, as can be seen in the temporal evolution of the carrier injection rate (**c**). The interference between these two events makes a dip appear in the normalized spectral response (**b**) in the same region where the measured spectral response has the minimum. The $\phi_{CE}$-controlled photoinjection events cause an abrupt increase of the energy deposited in the system. In **d**, we compare a highly nonlinear component of the energy, calculated using TDDFT, to the time integral of $E^8(t)$. The TDDFT calculations show a slight time delay of the effective carrier injection, $\tau_{inj}$, relative to the maximum of the laser field.

In the frequency domain, Eq. (2) simplifies to

$$S_d(\omega) = -A_d(\omega)G(\omega) = iE_d(\omega)G(\omega)/\omega. \qquad (4)$$

The dependence of the normalized spectrum $|S_d(\omega)|$ on $\phi_{CE}$ is shown in Fig. 3a. Using the ansatz $G(t) \propto E_i^8(t - \tau_{inj})$, we qualitatively reproduce the measured phase-dependent spectral response (Fig. 3b). In particular, both data sets exhibit a conspicuous minimum at frequencies near the carrier frequency of the injection pulse for $\phi_{CE} = \pi/2$, implying carrier injection in two similar bursts during the two central half cycles of $E_i$ (Fig. 3d). These two bursts yield opposing contributions to the Fourier transform of the gating function at the central frequency of the injection pulse thus leading to a pronounced minimum appearing in both $|G(\omega)|$ and $|S_d(\omega)|$ at $\omega_L$. Our predictions are not only found to be in agreement with our experimental observation displayed in Fig. 3a, but are also corroborated by the calculated energy exchange between the injection field and the medium in TDDFT calculations including a small temporal shift $\tau_{inj} \approx 75$ as, shown in Fig. 3d.

We thus find that the rate of carrier injection is modulated at twice the frequency of the laser on a sub-cycle basis, rather than following the cycle-averaged intensity of the pulse, consistent with the observations in thin solid films made with attosecond transient absorption measurements[3,15–17]. Furthermore, they

demonstrate that careful selection of $\phi_{CE}$ is required to confine the carrier injection to a single sub-femtosecond burst—a prerequisite for high-fidelity, PHz bandwidth waveform sampling.

**Attosecond timing of light-field-driven carrier motion.** For a gating function $G(t)$ consisting of a single sub-cycle spike centered near the maximum of the injection field, the measured signal would be proportional to the vector potential of the driving field. A photoinjection delay as predicted by the TDDFT simulations should shift the signal in time: $S_d(t) \propto A_d(t - \tau_{inj})$. We use EOS in the near-infrared spectral region[22,23] to directly relate the response of the field-induced currents to a known driving electric field, including exact timing relative to the injection field (see Supplementary Note 2 for details). In EOS, the presence of the drive field overlapping in time with the sampling pulse, in this case, $E_i$, results in a measurable change in the polarization state of $E_i$, which provides both the time-gated electric field of $E_d$ and the relative timing of $E_i$ and $E_d$ corresponding to each position of the time delay stage. Temporal offsets in the field vs. time delay measured by NPS and EOS thus provide insight into carrier dynamics inside of the NPS device that result in a delay of $G(t)$.

Employing a 12-μm-thick, type-II beta barium borate detection crystal, we performed high-frequency EOS[23] on an 1.8-μm pulse obtained through intrapulse difference-frequency generation,

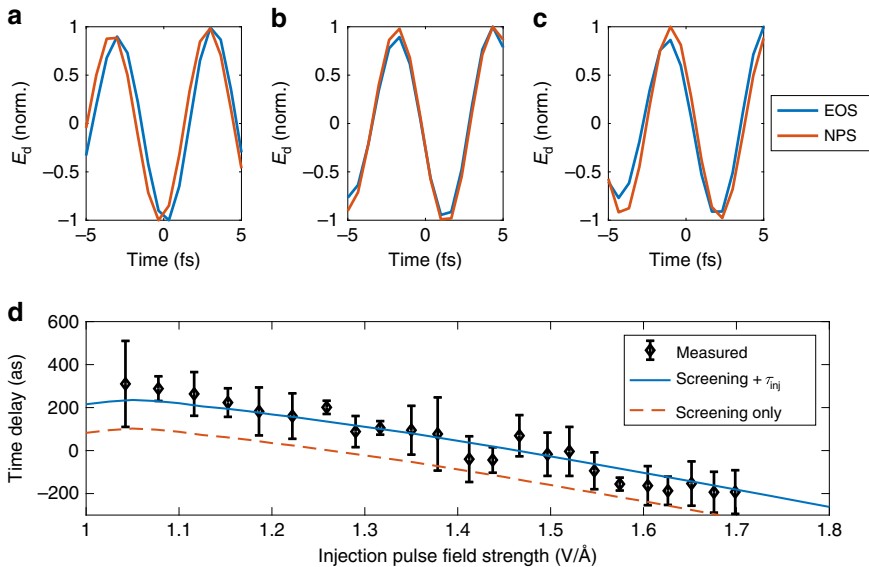

**Fig. 4 Attosecond timing of light-induced transitions in a solid via simultaneous EOS and NPS.** The two measured waveforms of the driving waveform $E_d$ exhibit a temporal shift due to the combination of delayed injection of charge carriers into the dielectric relative to the field maximum of the laser pulse and the influence of field screening by free carriers, shown as the relative shift of the waveforms at injection pulse field strengths of **a** 1.04 V Å$^{-1}$, **b** 1.4 V Å$^{-1}$, and **c** 1.7 V Å$^{-1}$. The time shift (**d**) depends on the field strength, decreasing as the amplitude of the injection field increases. The measured values are compared to expectations of electric field screening by the laser-induced free carriers, with and without a temporal delay fitted to the experimental data, $\tau_{inj} = 134 \pm 93$ as, comparable to the value of 75 as observed in the TDDFT calculations. Error bars represent the standard deviation of multiple measurements of the relative timing of the waveforms.

which both is detected via EOS and used as the driving field $E_d$ in NPS. The 750 nm, 2.7-fs pulse continues to be used as $E_i$ and functions as the sampling pulse for EOS. The pulse pair is reflected from a glass wedge translated into the beam before it reaches the NPS detector, reducing the intensity and redirecting the beam to the EOS crystal. Sequential measurements with the two techniques allow us not only to confirm the fidelity of the retrieved waveform of $E_d$ ($\rho = 0.93$, $\epsilon = 0.36$), but also fixes the relative timing of $E_i$ within the cycle of $E_d$ through the known temporal response of EOS.

The comparison between the EOS and NPS traces (Fig. 4) shows a temporal shift relative to the timing of the input injection pulse within the driving field which we relate to a delayed onset of the charge-carrier motion. This delay decreases with increasing field strength of the injection pulse in the range of peak electric fields >1 V Å$^{-1}$. Due to slight differences in the bandwidths of the two detectors, best agreement is expected near the center of the reconstructed pulse, where we measure the delay between EOS and NPS signals. Simulating the propagation of the injection and driving laser pulses into the material, we find that in addition to the intrinsic delay in the injection of free carriers $\tau_{inj}$, observed in the microscopic TDDFT simulations, the timing of the wave consists of two parts: a constant temporal offset and an intensity-dependent trend tied to the density of charge carriers induced in the medium. The observed trend is primarily due to a free-carrier-mediated cross phase modulation on the carrier-driving pulse, $E_d(t)$: the laser-injected carriers slightly reshape the electric field upon its propagation into the material (Supplementary Note 3 for details). A small number of carriers tends to reduce the refractive index, which causes the field present in the material to increase after carrier injection, but a further increase in the number of injected carriers leads to an increase in reflectivity and, in turn, causes the field to decrease after injection. The intensity-dependent decrease of the field at later times thus causes the apparent decrease of temporal delay through time-dependent reshaping of the waveform.

Access to a measurement technique of these subtle timing effects with ~100 as precision is essential for understanding how optical fields translate into electronic excitations in ultrafast circuits. Measurements of optical field effects *inside* of solid state devices without the need for the light or electrons to exit the material through surfaces or interfaces may also open a path to future investigation of the timing of ultrafast carrier injection. The physical origin of the measured and calculated injection delays $\tau_{inj}$ of 134 ± 93 and ≈75 as, respectively, are still not yet well understood.

**THz-to-PHz lightwave sampling.** Having established the suitability of NPS to detect the electric field $E_d(t)$ in the optical and near-infrared frequency ranges we also tested its frequency and sensitivity limits. To this end, we applied it to five waveforms with frequencies ranging from <60 THz (5.3 μm) to 1.1 PHz (275 nm), which exceeds 1 PHz continuous bandwidth. This collection of pulses was obtained using various methods for frequency conversion: difference-frequency generation and sum-frequency generation under two different phase-matching conditions. The waveforms measured by NPS and the dependence of the NPS signal on the peak power of the driving pulse are shown in Fig. 5. The analysis of NPS measurements based on Eq. (2) requires that the medium response to the driving field be linear, which implies linear scaling of the signal with respect to the peak power. Ranges of the peak driving-pulse power that satisfy linearity requirements can be inferred from Fig. 5b. To our knowledge, this is the only solid-state field-detection technique to date to have demonstrated such a wide detection bandwidth. The technique shows a dynamic range up to 30 dB, where variations between the waveforms arise from differences in the focusability of the nonlinearly generated light beams, the noise level being higher for shorter wavelengths, and the fact that, for a given amplitude of the electric field, the amplitude of $A_d$ is proportional to $\lambda_L$. The lowest pulse energies measured, on the nJ level, would produce an

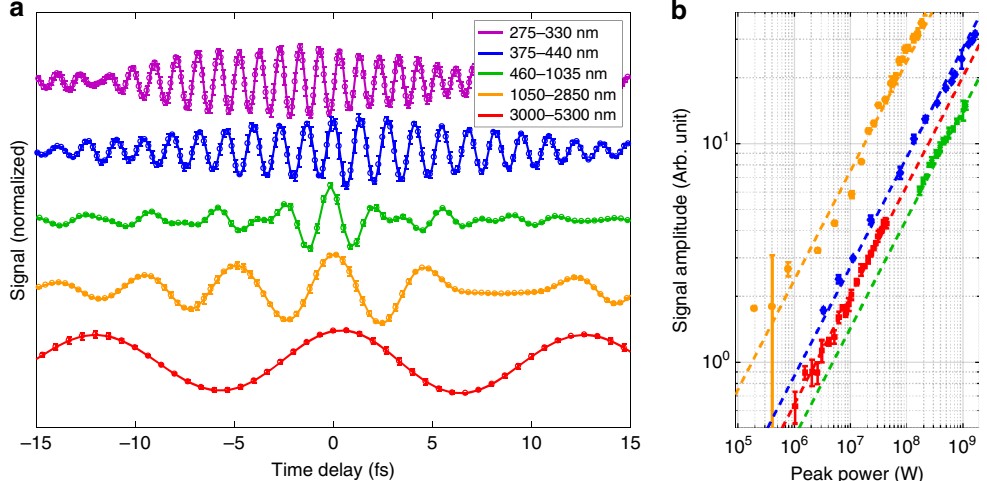

**Fig. 5 Broadband electric-field measurement from the infrared to ultraviolet. a** Waveforms in different spectral ranges spanning the mid-infrared to ultraviolet are detected by NPS, triggered by a 2.7 fs near-infrared visible pulse. The waveforms covering 3000–5300 and 1050–2850 nm are generated via difference-frequency generation in LiNbO$_3$ and BBO, respectively. The waveform covering 460–1035 nm is generated by the spectral broadening of a 21 fs, 790 nm pulse from a Ti:sapphire laser in a neon-filled hollow-core fiber and compressed with chirped mirrors to 2.7 fs duration (this waveform drives all other nonlinear processes presented). The waveforms covering the spectral ranges 375–440 and 275–330 nm are both generated via sum-frequency generation in BBO, using different phase-matching conditions. The error bars represent the standard deviation of repeated measurements. **b** Using variable attenuation, the driving pulse energies are reduced and the signal is plotted vs. input pulse peak power, exhibiting both the linearity of the field response and its large dynamic range. The dashed lines are fitted linear field responses (i.e. scaling with $\sqrt{P}$, the square root of the peak power). The absolute positions of the lines on the vertical scale result from both the wavelength-dependent sensitivity of the measurement, which favors longer wavelengths, and the spatio-temporal properties of the frequency-converted beams. This causes the driving field strength, which determines the signal strength, to depend on more variables than the power alone, but their proportionality with fixed beam parameters exhibits the linearity of the measurement.

energy shift of only a few meV in a typical attosecond streaking measurement, requiring many hours of measurement time to be reliably detected, in a significantly larger and technically complex experimental infrastructure.

## Discussion

We have measured the waveforms of optical fields from infrared to ultraviolet frequencies. Comparison between measured and simulated data gives access to the timing and temporal evolution of light-induced transitions in a solid with sub-femtosecond accuracy and resolution. These carrier injection dynamics in wide band-gap solids are now measurable on their natural timescale *inside bulk solids*, inaccessible by attosecond XUV pulses, thus providing unique insight into the interaction responsible for fundamental processes in optoelectronics. The demonstrated petahertz-bandwidth waveform recorder has a sensitivity sufficient to measure the oscillating field of light pulses generated directly from laser oscillators and nonlinear frequency converters driven by them.

Few-cycle infrared-to-ultraviolet light pulses fully-characterized by this solid-state instrumentation constitute a direct probe of electron dynamics in the valence and conduction band of solid-state materials. Attosecond metrology of the laser electric field and the material polarization induced by this field has the potential to become as simple and cost-effective as femtosecond metrology. Femtoscience made a vast array of contributions to fundamental research in physics, chemistry, and biology as it joined the investigative arsenal of laboratories around the world. Analogously, the emerging toolbox of all-solid-state attosecond metrology now promises a similar proliferation, especially with regard to electronic motion.

## Methods

**Optical system**. Pulses from a CEP-stabilized, Kerr lens mode-locked Ti:Sapphire oscillator (Femtolasers Rainbow 2) are amplified to 1 mJ pulse energy at 3 kHz repetition rate in a Ti:Sapphire multipass amplifier (Femtolasers Femtopower). The

pulses are then spectrally broadened in a 1.1 m long, neon-filled hollow capillary and compressed using custom chirped mirrors with a bandwidth supporting 500–950 nm.

The injection field $E_i$ is obtained by taking the reflection from the surface of a fused silica wedge, after which the pulse passes through a second wedge pair for dispersion control, a delay line mounted on a piezoelectric translation stage, and a wiregrid polarizer (LOT-QuantumDesign).

The pulse responsible for producing the driving field $E_d$ passes through the wedge pair from which $E_i$ was reflected, then through a pair of wiregrid polarizers for amplitude control. The light is then transmitted through a nonlinear crystal to create waveforms of various central wavelength (Figs. 4 and 5 in the main text).

The two fields $E_i$ and $E_d$ are recombined with orthogonal polarization on a wiregrid polarizer, the former being transmitted and the latter reflected. The pulses are then focused using an off-axis parabolic mirror onto the sample, to a beam spot smaller than the gap between the gold electrodes.

Addition information describing the measurement, noise, and artifacts can be found in Supplementary Notes 6–11.

**Signal acquisition**. The signal from the electrodes is first amplified in a transimpedance amplifier and then through a lock-in amplifier, referenced either to a carrier-envelope phase modulation, or an amplitude modulation generated on the drive waveform by an optical chopper. The acquired signal is bandpass filtered as described in Supplementary Note 1.

## Data availability

The raw data that support the findings of this study are available from the corresponding author upon reasonable request.

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

## Acknowledgements

This research is based upon work supported by the US Air Force Office of Scientific Research under award number FA9550-16-1-0073. This work was supported by the FWF Austria (SFB-041 ViCoM, SFB-049 NextLite and doctoral college W1243), the COST Action CM1204 (XLIC), and the IMPRS-APS. Calculations were performed using the Vienna Scientific Cluster (VSC).

## Author contributions

N.K. and F.K. initiated and conceived the study. S.S., D.Z., and S.K. performed the NPS and EOS measurements supervised by N.K. F.S. performed the attosecond streaking measurements supervised by M.S. I.F., C.L., and J.B. performed the TDDFT calculations and relevant interpretation thereof. M.W. and V.S.Y. provided additional theoretical interpretation. S.S. and N.K. performed the data analysis. N.K., M.S., and F.K. wrote the manuscript with input from all authors. All authors discussed the results.

## Competing interests

The authors declare no competing interests.
