## [Peer Review File · Nature Communications]

Editorial Note: This manuscript has been previously reviewed at another journal that is not operating a transparent peer review scheme. This document only contains reviewer comments and rebuttal letters for versions considered at *Nature Communications*. Mentions of the other journal have been redacted.

Reviewers' comments:

Reviewer #1 (Remarks to the Author):

NCOMMS-19-18655-T - Attosecond optoelectronic field measurement in solids

Sederberg et al

The authors present a new method to characterise "optical" waveforms from the ultraviolet to infrared spectral regions based on strong-field nonlinear photoabsorption in solid dielectrics: nonlinear photoconductive sampling (NPS). The author verify the method via a comparison to a existing waveform characterization methods, namely attosecond streaking (AS) for a near single-cycle pulse centred at 750nm and electro-optic sampling (EOS) for a multi-cycle pulse centred at 1.8 μ m. The authors also compare the relative timing of measurements made between NPS and EOS to determine the effective delay induced by the photoabsorption to compare to and verify a model of the process. Compared to previously demonstrated waveform characterization methods, NPS is extremely simple merely requiring a polarization interferometer suitable for single-cycle pulses followed by focusing onto a solid-state device. Although the setup is extremely simple, the accuracy of the results and the limitations placed on the sampling pulse reduce the potential impact.

The authors have made a solid effort to address the concerns of the referees, and have significantly improved the manuscript as a result, although there are still some issues that I have.

Regarding the waveform fidelity (particularly the comparison between AS and NPS). I calculate a substantially different number for the RMS field error, namely 0.34 compared to their 12%. Could the authors please provide a formula or a reference for their calculation. Note that the RMS field error described by Dorrer as previously suggested is technically not a percentage error since it varies from 0 to 2. However a value of 2 corresponds to two fields that are inverses of each other, so this value can be considered as a percentage error directly, although this was not explicitly stated in the paper by Dorrer -

hence the authors should be clear. Regarding a the Pearson correlation - I calculate a similar number of 0.94 compared to the authors 0.88, and hence I expect an error in the calculation of the RMS error - have the authors missed the square root: $0.34^2 \sim 0.117$. I doubt such a large discrepancy can be put down to the truncated waveforms used in my calculations (data extracted from the paper) because the contribution to the error is an order of magnitude lower outside the plotted region due to the reduced intensity. Since the two fields are unlikely to match perfectly outside the plotted region, it is reasonable to expect the correlation and RMS errors to decrease as the calculation window increases.

One should note that these numbers, whilst extremely useful to unambiguously quantify the difference between the two fields, are global quantities. Indeed, I do not dispute that the measurements do measure pulses at the same central wavelength with a similar temporal intensity distribution - although as previously pointed out, even that difference is not insignificant. However, the purpose of the method is to measure the waveform, and there are significant differences between the two. The impact of the method is only sufficiently strong for Nature Communications if it is possible to accurately characterise the "sub-cycle" nature of the waveform since the method can then be used to study field-sensitive processes such as the correlated electron dynamics induced in condensed matter by strong laser fields (particularly with photon energies well below the band-gap) and other strong-field phenomenon such as high harmonic generation and above threshold ionization.

The authors have added a section to the SI on the harmonic artefact which provides reasonable evidence of its effects. This still leaves open the question on the significant difference between measured waveforms and corresponding spectra between the AS and NPS measurements.

The authors present experimental information on the signal to noise ratio in terms of absolute field values - which is a reasonable approach from a practical/experimentalist point of view. The authors need to be consistent with their units since it is difficult to compare the field strength used in the SNR measurements (V/A) to the pulse energy used in the detection limits (μJ or nJ). This leaves some confusion as to the sensitivity of the method, since the authors claim an optimal range of peak driving field strength of 0.2-0.8V/A, and the graph shows a SNR of 1000 @ $E_d \sim 0.1\text{V/A}$.

The authors do not present any statistical information on the precision of the measurement - for example how repeatable are the measurements (e.g. some kind of density plot of multiple measurements of the waveforms or equivalent) and how accurately are known changes in the waveform recovered (e.g. measurement of known dispersion/ & CEP shifts)? Such evidence is common practice in pulse metrology.

The authors address the issue of spectral response in the supplementary information. Whilst this is correct, I found it a little difficult to follow and therefore recommend re-wording. In addition, the final result, namely the maximum pulse duration ~ 3.8 fs at 800nm, should be quoted in the main text as a reasonable upper limit on the pulse duration necessary to provide a reasonably accurate reconstruction without the need to apply any deconvolution. Whilst such a pulse is relatively easily obtained, it still represents a tight limit, requiring state of the art but readily available systems and careful pulse management to achieve, thus reducing the potential impact.

The authors claim that a deconvolution of the spectral response is possible if the injection field is known. I have reservations about such claims: (1) deconvolution is notoriously difficult to achieve well for experimental data due to the effects of noise - simply applying a bandpass filter is not sufficient; and (2) one should not expect to have to characterise the pulse used to sample the waveform to be able to use the device. Even in this case, the precision of such characterisation methods are typically not fantastic in practice, so again affecting the fidelity of the reconstruction.

The expression for the spectral response is also a little complex - it would be nicer to obtain a much simpler expression in the limit of a short pulse.

The authors have significantly added to the details of the EOS measurements. Their simulations suggest a nonlinear contribution to the EOS time delay of ~ 25 as, and claim a screening delay of 134 ± 93 as (measured) and 75as (simulated). The authors should use a statistical measure (e.g. t-test / % confidence etc.) of the significance of the difference between the measured screening delay and simulated screening delay, and compare that to the significance of the measured screening delay to the nonlinear EOS delay. A simple comparison of the difference relative to the standard error is not sufficient to quantify the significance of the difference. So far the authors have not so far provided sufficient statistical analysis as evidence to reject the null hypothesis that the delay is not caused by nonlinear/propagation effects in EOS.

Whilst the authors do indeed state that the EOS sampling is taken as reflection from wedges, this does not make the setup explicitly clear: what is the field strength of the pulses in the EOS setup; what is the relative timing delay between the driving and injecting fields introduced by the wedges and with what accuracy is this known; is the intensity/field strength of the EOS measurements held fixed as the intensity/field strength used in the NPS measurements is varied? I go back to my original point that the authors are claiming a measured delay on the order of 100as using a field with period of ~ 6 fs, corresponding to $\sim 1.5\%$ of a period. Given that the nonlinear delay is a significant fraction of the

screening delay ($\sim 25\text{as}$ compared to $\sim 75\text{as}$), more detailed statistical and systematic error analysis needs to be performed to be able to attribute any significance to one delay over the other.

Overall, I believe that the authors have demonstrated a potentially simple waveform characterisation method that works in a restricted but reasonably accessible regime with at best mediocre accuracy. Whilst I genuinely believe the method and understanding to be a step forward, I still have doubts about the robustness of the results and the analysis used and therefore struggle to believe the impact is high enough for Nature Communications.

I also have some minor comments about the manuscript:

MAIN ARTICLE:

Please be consistent with the units used. From a physical point of view, it is the field strength that matters, and therefore this number needs to be quoted. However, from a practical point of view, it is the pulse energy that is measured, along with the known pulse duration and spot size. I therefore recommend quoting both in the manuscript.

I disagree with the statement "We attribute a significant portion of the differences between the retrieved fields to different pulse distortions in the different reflective and transmissive optical elements in the two measurement systems." (P4).

The statement "... inaccessible by attosecond XUV/X-ray pulses" (P8, conclusion) does not hold for X-ray pulses - since these will transmit and interact with the bulk.

The statement "... most fundamental processes in optoelectronics." (P8, conclusion) is subjective - remove or replace most with some (which means at least 1, possibly all).

The statement "... constitute a uniquely direct probe of electron dynamics ..." (P8, conclusion) has no evidence - remove uniquely.

Final statement: "As femtoscience has made a vast array of contributions to fundamental research in physics, chemistry and biology as it has joined the investigative arsenal of laboratories around the world, the emerging toolbox of all-solid-state attosecond metrology now holds promise for a similar proliferation, especially with regard to electronic motion" is not quite grammatically clear, and potentially overselling - I do not believe the authors have demonstrated with sufficient evidence that all-solid-state attosecond metrology holds sufficient accuracy for the latter statement to hold.

Add limits of pulse duration (i.e. 3.8fs FWHM at 800nm central wavelength).

SUPPLEMENTARY INFORMATION

P3: "For the sake of numerical stability, the first two terms in the Taylor series expansion in P are utilized in the following" - does not make sense: how do the first two terms aid numerical stability?

P3: " the polarization due to the laser-induced charge carriers both depends on and influences the transmitted field and the two must be solved self-consistently." - does not make sense, what does solved self-consistently mean?

P4: Acronym TDDFT used without prior definition.

P13: "One may note several features in the spectrum of the driving laser pulse in Fig. S10(b). The NPS measurement reproduces (within the limits of the spectral resolution) the structure of the spectrum, with the exception of the region above 600 THz (500 nm)." There appears to be a discrepancy in (c) below 0.35THz and the NPS and AS spectral do not agree across the bulk of the bandwidth. The central wavelengths of the longer pulses also seem offset.

P13-14: "The waveforms shown in Figure 4 are zoomed into the maximum of the envelope of the electric field, where the most reliable timing information exists. Figure S11 shows a wider view of the pulses measured by the two detection techniques. The agreement between the waveforms is quantitatively better than the comparison with attosecond streaking, but there are still differences visible in the traces at 13th the rising and falling edges of the pulse. This is due to the slightly different

effective bandwidths of the measurements, which, in the case of a compressed pulse, lead to differences in the waveform proportional to the gradient of its envelope. For this reason, the timing comparison is done using the cycle of the waveform at the maximum of the envelope, where the time derivative is minimized." This is a very subjective measure of delay, very prone to error and also a badly posed problem. How can one compare the delay between two pulses that have different shapes. Indeed, when the pulse is a simple bell-shaped curve (e.g. Gaussian), such a question may seem reasonable since the outputs are also simple bell shaped curves. However, this can be misleading since is it a genuine time-delay or a simple asymmetric temporal response - they are not the same if one considers a structured pulse. Comparison of the field between the EOS and NPS signals suggest a wavelength difference that results in phase shift/timing variation in the waveform oscillations that varies by an amount over the waveform that is comparable to the screening delay. To be explicit: at $t \sim 20$ fs, the EOS signal precedes the NPS signal, but at $t \sim 20$ fs the EOS signal proceeds the NPS signal.

Reviewer #2 (Remarks to the Author):

Dear authors

I sincerely submit my review report as an attached file.

The manuscript written by Sederberg et al. is the revised version which is transformed to Nature Communications from the previous submission to [REDACTED] [REDACTED]. Again, the development of direct sampling techniques of an optical field or a light waveform with petahertz bandwidth is one of the most important and broad-interest topics in not only the attosecond technology but also the broad optical science fields. The topic presented in this paper, therefore, matches the current scope of Nature Communications. Although several concepts for direct sampling techniques of an optical field, including attosecond streaking (ref. 26), petahertz oscilloscope (ref. 28), attosecond sampling of arbitrary optical waveforms (ref. 27), the tunneling ionization with a perturbation for the time-domain observation of an electric field (ref. 30), and the optical-field solid state detector (ref. 1), were demonstrated so far, their measurement results have not been evaluated by a comparison with another demonstrated method. Generally, it is significantly necessary to compare different methods for establishing new technology as reliable one. However,

the comparison between different methods has been difficult task due to the experimental difficulty for attosecond temporal measurement setups.

In the previous revision, I pointed out the following three points as the new findings which Sederberg et al. concluded in this manuscript:

- (1) The ability to retrieve the electric field waveform of light from a simple current measurement in a solid state device.
- (2) Its unprecedented bandwidth of detection.
- (3) The ability to determine the temporal evolution of the charge carrier density from the measurement.

I made a reservation for [REDACTED] publication because I could not help but judged the item (1), which was the main argument in the previous manuscript, less impact compared with the first demonstration in Schiffrin et al. Nature 493, 70 (2013).

In the current revised version for Nature Communications, the authors sincerely revised my major concern. Especially, they add a clear discussion comparing the new main finding in the current work with the previous related works of refs. 1, 36, 37, in the middle part of the “Optical field sampling in solids” section in page 4. In this discussion, they clearly describe that they test and verify the relationship between the measured signal of the induced current and the driving field by comparing results of their NPS measurements to those of attosecond streaking and electro-optical sampling, which were well established methods of direct measurements of optical field oscillations. Although each technique, which corresponds to NPS, attosecond streaking, and electro-optical sampling, is previously demonstrated in ref. 1, ref. 38, and ref. 23, respectively, the combination of all the techniques is the first and valuable experimental demonstration for strengthening the functionality of the NPS method. In addition, the authors applied the NPS technique to five waveforms with frequencies ranging from <60 THz (5300 nm) to 1.1 PHz (275 nm) for demonstrating its unprecedented bandwidth of optical field sampling measurement. Such measurement, investigating the detection bandwidth of proposed method, is also the first demonstration in the direct optical-field sampling techniques reported previously. From this point of view, the manuscript is an important extension work of refs. 1, 36, and 37, which were reported by the authors’ related group of MPQ, in the attosecond science field community, while the technological concept of the detection scheme presented in the current manuscript cannot be said totally new.

In the following, minor comments are described.

- (1) The authors clearly describes the current signal, $S_d(\tau)$, as the convolution between the vector potential $A_d(t)$ and the gating function $G_d(t-\tau)$ in the current revised version. And also they comment that the noise level being higher for shorter wavelengths can be explained by the fact that, for a given amplitude of the electric field, the amplitude of $A_d(t)$ is proportional to the wavelength of the driving field. Although this description is reasonable, the signal amplitude $S_d(\tau)$ does not show the wavelength dependence at fixed power in Fig. 5(b). The author should comment on that shortly.

(2) In addition to the above comment related to Fig. 5(b), the slopes of the signal as a function of driving laser power for several wavelengths seems almost same. Does the linearity of this curves originate from the simple Ohm's law in the high-frequency region? Under the assumption of simple Ohm's law, the same slope indicates the same resistance for the electronic transport at wide range of frequency from 60 THz to 1 PHz. Does this indicates the scattering process of electrons driven by optical field shows no frequency dependence? It seems that such finding should be pointed out and discussed by the authors.

In summary, I find that their description in the current manuscript does contain the sufficient novelty to justify a publication in Nature Communications.

Reviewer #4 (Remarks to the Author):

Dear authors,

I want to apologize for the delay to submit my review. Even though the nonlinear photoconductive sampling approach was published before by your group, I consider that the manuscript has sufficient novel results to deserve publication in Nature Communications. Here are the reasons to support my decision:

- First, I am very impressed by the measured delay between NPS and EOS as a function of the field strength of the photoexcitation pulses. This brings novel physics compared with previous published work.

- Second, the results of NPS from the UV to the mid-infrared are very impressive and denotes a technological breakthrough in optical science.

Regarding the comparison of NPS and EOS measurements, and the time delay as a function of field strength, I was disappointed by the short discussion of the manuscript. I consider this as an important physics milestone and the authors should better explained their results and even speculated on the origin of the delay. From the TDDFT, the authors have found an intrinsic delay. What is the physical origin of this intrinsic delay? Second, the authors are discussing in the manuscript that the additional delay comes from cross-phase modulation. I am surprised that increasing the field strength reduces the delay. The authors should clarify this paragraph on page 7.

Regarding the NPS measurements from the UV to the mid-infrared (Figure 5), I am surprised that the waveforms are looking like continuous functions, as compared to the results presented in Figure 4. Also, since those measurements arise from pump-probe scans, I expect that several scans have been acquired. It will be nice to see the repeatability of the measurements and some error bars, to judge the validity of the technique.

Reviewer #1 (Remarks to the Author):

NCOMMS-19-18655-T - Attosecond optoelectronic field measurement in solids
Sederberg et al

The authors present a new method to characterise "optical" waveforms from the ultraviolet to infrared spectral regions based on strong-field nonlinear photoabsorption in solid dielectrics: nonlinear photoconductive sampling (NPS). The author verify the method via a comparison to a existing waveform characterization methods, namely attosecond streaking (AS) for a near single-cycle pulse centred at 750nm and electro-optic sampling (EOS) for a multi-cycle pulse centred at 1.8um. The authors also compare the relative timing of measurements made between NPS and EOS to determine the effective delay induced by the photoabsorption to compare to and verify a model of the process. Compared to previously demonstrated waveform characterization methods, NPS is extremely simple merely requiring a polarization interferometer suitable for single-cycle pulses followed by focusing onto a solid-state device. Although the setup is extremely simple, the accuracy of the results and the limitations placed on the sampling pulse reduce the potential impact.

The authors have made a solid effort to address the concerns of the referees, and have significantly improved the manuscript as a result, although there are still some issues that I have.

Regarding the waveform fidelity (particularly the comparison between AS and NPS). I calculate a substantially different number for the RMS field error, namely 0.34 compared to their 12%. Could the authors please provide a formula or a reference for their calculation. Note that the RMS field error described by Dorrer as previously suggested is technically not a percentage error since it varies from 0 to 2. However a value of 2 corresponds to two fields that are inverses of each other, so this value can be considered as a percentage error directly, although this was not explicitly stated in the paper by Dorrer - hence the authors should be clear. Regarding a the Pearson correlation - I calculate a similar number of 0.94 compared to the authors 0.88, and hence I expect an error in the calculation of the RMS error - have the authors missed the square root: $0.34^2 \sim 0.117$. I doubt such a large discrepancy can be put down to the truncated waveforms used in my calculations (data extracted from the paper) because the contribution to the error is an order of magnitude lower outside the plotted region due to the reduced intensity. Since the two fields are unlikely to match perfectly outside the plotted region, it is reasonable to expect the correlation and RMS errors to decrease as the calculation window increases.

We calculated the RMS error as the standard root mean square of the difference between the normalized waveforms, and have reconfirmed that our calculation of this value was correct. The explicit formula we have used is as follows:

$$\text{Err}_{\text{RMS}} = \sqrt{\frac{\sum_{n=1}^N (E_{\text{NPS}}(t_n) - E_{\text{streaking}}(t_n))^2}{N}}$$

where N is the total number of sampled points contributing to the waveforms.

One should note that these numbers, whilst extremely useful to unambiguously quantify the difference between the two fields, are global quantities. Indeed, I do not dispute that the measurements do measure pulses at the same central wavelength with a similar temporal intensity distribution - although as previously pointed out, even that difference is not insignificant. However, the purpose of the method is to measure the waveform, and there are significant differences between the two. The impact of the method is only sufficiently strong for Nature Communications if it is possible to accurately characterise the "sub-cycle" nature of the waveform since the method can then be used to study field-sensitive processes such as the correlated electron dynamics induced in condensed matter by strong laser fields (particularly with photon energies well below the band-gap) and other strong-field phenomenon such as high harmonic generation and above threshold ionization.

The authors have added a section to the SI on the harmonic artefact which provides reasonable evidence of its effects. This still leaves open the question on the significant difference between measured waveforms and corresponding spectra between the AS and NPS measurements

To the best of our knowledge, the difference between the waveforms is primarily due to the differences in beam path and focusing conditions of the two measurements. In view of the two different optical pathways to the AS and NPS detectors we do not think that the difference between retrieved waveforms should be taken an indication of shortcomings of NPS but, instead, most likely reflects the existing differences in waveform at different points of the measurement.

The authors present experimental information on the signal to noise ratio in terms of absolute field values - which is a reasonable approach from a practical/experimentalist point of view. The authors need to be consistent with their units since it is difficult to compare the field strength used in the SNR measurements (V/A) to the pulse energy used in the detection limits (uJ or nJ). This leaves some confusion as to the sensitivity of the method, since the authors claim an optimal range of peak driving field strength of 0.2-0.8V/A, and the graph shows a SNR of 1000 @ $E_d \sim 0.1V/A$.

The authors do not present any statistical information on the precision of the measurement - for example how repeatable are the measurements (e.g. some kind of density plot of multiple measurements of the waveforms or equivalent) and how accurately are known changes in the waveform recovered (e.g. measurement of known dispersion/ & CEP shifts)? Such evidence is common practice in pulse metrology.

The repeatability of the delay measurements is represented by the error bars in Figure 4: each measurement was repeated multiple times, the error bars on individual points represent the standard deviation of the individual measurements in the set represented by the point. Error bars have also been added now to Figure 5, showing the standard deviation of repeated measurements of the same waveform in all spectral ranges - they are quite small as the measurements exhibit a high degree of repeatability.

We indeed have already performed the analysis of the spectral phase of a pulse going through a known thickness of material and verified that it is consistent with the refractive index known from literature. We have added a set of this data to the SI, section 11.

The authors address the issue of spectral response in the supplementary information. Whilst this is correct, I found it a little difficult to follow and therefore recommend re-wording. In addition, the final

result, namely the maximum pulse duration ~ 3.8 fs at 800nm, should be quoted in the main text as a reasonable upper limit on the pulse duration necessary to provide a reasonably accurate reconstruction without the need to apply any deconvolution. Whilst such a pulse is relatively easily obtained, it still represents a tight limit, requiring state of the art but readily available systems and careful pulse management to achieve, thus reducing the potential impact.

We agree that the pulses must be short for the method to work, but also that a large number of labs around the world have such pulses available (and those working with short pulses like these are in need of additional methods of characterization, such as NPS will offer). We considered the exact pulse parameters required to be a purely technical matter better left for the SI, but have inserted the number into the main text.

The authors claim that a deconvolution of the spectral response is possible if the injection field is known. I have reservations about such claims: (1) deconvolution is notoriously difficult to achieve well for experimental data due to the effects of noise - simply applying a bandpass filter is not sufficient; and (2) one should not expect to have to characterise the pulse used to sample the waveform to be able to use the device. Even in this case, the precision of such characterisation methods are typically not fantastic in practice, so again affecting the fidelity of the reconstruction.

The expression for the spectral response is also a little complex - it would be nicer to obtain a much simpler expression in the limit of a short pulse.

We agree that deconvolution can only be performed when attention is paid to the noise level, and to how consistent the signal to noise level is across the measurement. However, we are not suggesting a blind deconvolution: rather to first know the form of the probe pulse. This allows the factor by which different frequency components must have their amplitudes increased prior to performing the analysis, which can be compared to the signal-to-noise level of the experiment to know a priori if deconvolution is appropriate for the measurement, and in which frequency range.

The limit in the case of a short pulse is simply a flat spectral response $G(\omega) = \text{const.}$, where $w(t)$ becomes a delta function and the recorded waveform is $A(t)$. We have added a comment to this effect to the relevant section in the SI.

The authors have significantly added to the details of the EOS measurements. Their simulations suggest a nonlinear contribution to the EOS time delay of ~ 25 as, and claim a screening delay of 134 ± 93 as (measured) and 75as (simulated). The authors should use a statistical measure (e.g. t-test / % confidence etc.) of the significance of the difference between the measured screening delay and simulated screening delay, and compare that to the significance of the measured screening delay to the nonlinear EOS delay. A simple comparison of the difference relative to the standard error is not sufficient to quantify the significance of the difference. So far the authors have not so far provided sufficient statistical analysis as evidence to reject the null hypothesis that the delay is not caused by nonlinear/propagation effects in EOS.

The error in the 134 ± 93 as value was obtained via the confidence interval returned through the statistical analysis of the data that had already been performed, which resulted from the calculated Jacobian matrix describing the derivative of the error with delay and the residuals, which is straightforwardly reduced to the standard error. The 95% confidence interval is 115 to 195 as. However, the 91 as experimental timing jitter noise must be considered and added in quadrature, yielding the stated uncertainty in the value. The EOS-related delay is *maximally* 25 as in the intensity range tested. It varies linearly with intensity and is not a constant offset of the type being discussed; it cannot describe the observed effect, and would rather appear a distortion of the carrier-induced cross-phase modulation.

Whilst the authors do indeed state that the EOS sampling is taken as reflection from wedges, this does not make the setup explicitly clear: what is the field strength of the pulses in the EOS setup; what is the relative timing delay between the driving and injecting fields introduced by the wedges and with what accuracy is this known; is the intensity/field strength of the EOS measurements held fixed as the intensity/field strength used in the NPS measurements is varied? I go back to my original point that the authors are claiming a measured delay on the order of 100as using a field with period of ~ 6 fs, corresponding to $\sim 1.5\%$ of a period. Given that the nonlinear delay is a significant fraction of the screening delay (~ 25 as compared to ~ 75 as), more detailed statistical and systematic error analysis needs to be performed to be able to attribute any significance to one delay over the other.

Reflection from a glass surface does not induce a timing delay between the pulses. This can be verified by the Fresnel equations (the reflection coefficients are real-valued in this case). As stated in the manuscript, to perform the EOS measurement, the two pulses reflect from the surface of a glass wedge, sending them to the EOS crystal rather than the NPS detector.

Overall, I believe that the authors have demonstrated a potentially simple waveform characterisation method that works in a restricted but reasonably accessible regime with at best mediocre accuracy. Whilst I genuinely believe the method and understanding to be a step forward, I still have doubts about the robustness of the results and the analysis used and therefore struggle to believe the impact is high enough for Nature Communications.

We appreciate that the reviewer recognizes the importance of this measurement technique.

I also have some minor comments about the manuscript:

MAIN ARTICLE:

Please be consistent with the units used. From a physical point of view, it is the field strength that matters, and therefore this number needs to be quoted. However, from a practical point of view, it is the pulse energy that is measured, along with the known pulse duration and spot size. I therefore recommend quoting both in the manuscript.

I disagree with the statement "We attribute a significant portion of the differences between the retrieved fields to different pulse distortions in the different reflective and transmissive optical elements in the two measurement systems." (P4).

We assume that this comment is related to wording. We have rephrased it to: "The two measurement systems include different focusing and beam transport optics, to which we believe a significant amount of the differences between the fields may be attributed."

The statement "... inaccessible by attosecond XUV/X-ray pulses" (P8, conclusion) does not hold for X-ray pulses - since these will transmit and interact with the bulk.

We have removed the "X-ray" part of this statement - attosecond XUV pulses experience very high loss in materials such as silicon dioxide. The fact that sub-micron samples were required for previous attosecond studies that took place in our lab was indeed a driving factor behind this work.

The statement "... most fundamental processes in optoelectronics." (P8, conclusion) is subjective - remove or replace most with some (which means at least 1, possibly all).

We have removed the word "most".

The statement "... constitute a uniquely direct probe of electron dynamics ..." (P8, conclusion) has no evidence - remove uniquely.

We have removed the word "uniquely".

Final statement: "As femtoscience has made a vast array of contributions to fundamental research in physics, chemistry and biology as it has joined the investigative arsenal of laboratories around the world, the emerging toolbox of all-solid-state attosecond metrology now holds promise for a similar proliferation, especially with regard to electronic motion" is not quite grammatically clear, and potentially overselling - I do not believe the authors have demonstrated with sufficient evidence that all-solid-state attosecond metrology holds sufficient accuracy for the latter statement to hold.

We have rephrased this sentence.

Add limits of pulse duration (i.e. 3.8fs FWHM at 800nm central wavelength).

We refer the referee to our answer above and to the statement about the requirements to the pump pulse added to the text. However, as the maximum pulse duration is not a limiting factor for state-of-the-art laser labs we do not think that this technical detail is needed in the conclusions of our paper.

SUPPLEMENTARY INFORMATION

P3: "For the sake of numerical stability, the first two terms in the Taylor series expansion in P are utilized in the following" - does not make sense: how do the first two terms aid numerical stability?

The truncated Taylor expansion avoids errors in the numerical analysis that can occur when the field and polarization are out of phase, now also mentioned in the SI.

P3: " the polarization due to the laser-induced charge carriers both depends on and influences the transmitted field and the two must be solved self-consistently." - does not make sense, what does solved self-consistently mean?

Self consistent means that, since the field influences the carrier density, and the carrier density influences the field, care must be taken (self-consistent field cycle iterations) to ensure that these aspects of the dynamics are consistent.

P4: Acronym TDDFT used without prior definition.

Definition added.

P13: "One may note several features in the spectrum of the driving laser pulse in Fig. S10(b). The NPS measurement reproduces (within the limits of the spectral resolution) the structure of the spectrum, with the exception of the region above 600 THz (500 nm)." There appears to be a discrepancy in (c) below 0.35THz and the NPS and AS spectral do not agree across the bulk of the bandwidth. The central wavelengths of the longer pulses also seem offset.

Differences in spectral weight are not in general surprising, even if we would prefer perfect overlap: NPS measures the spectral distribution in the focus (with some time gating applied due to the finite delay scan), while the spectrometer measures the average spectrum of the beam. Spatiotemporal distortions produce differences.

P13-14: "The waveforms shown in Figure 4 are zoomed into the maximum of the envelope of the electric field, where the most reliable timing information exists. Figure S11 shows a wider view of the pulses measured by the two detection techniques. The agreement between the waveforms is quantitatively better than the comparison with attosecond streaking, but there are still differences visible in the traces at 13th rising and falling edges of the pulse. This is due to the slightly different effective bandwidths of the measurements, which, in the case of a compressed pulse, lead to differences in the waveform proportional to the gradient of its envelope. For this reason, the timing comparison is done using the cycle of the waveform at the maximum of the envelope, where the time derivative is minimized." This is a very subjective measure of delay, very prone to error and also a badly posed problem. How can one compare the delay between two pulses that have different shapes. Indeed, when the pulse is a simple bell-shaped curve (e.g. Gaussian), such a question may seem reasonable since the outputs are also simple bell shaped curves. However, this can be misleading since is it a genuine time-delay or a simple asymmetric temporal response - they are not the same if one considers a structured pulse. Comparison of the field between the EOS and NPS signals suggest a wavelength difference that results in phase shift/timing variation in the waveform oscillations that varies by an amount over the waveform that is comparable to the screening delay. To be explicit: at $t \sim 20$ fs, the EOS signal precedes the NPS signal, but at $t \sim 20$ fs the EOS signal proceeds the NPS signal.

It is exactly for this reason that the comparison at the central cycle is used. It is robust against these differences in spectral response due to the fact that the time derivative of the envelope is minimized. As the reviewer described, the differences are confined to the region of high temporal derivative of the envelope, which is why temporal delay analysis

is more accurate near the maximum. The reason that the analysis is done this way is not subjective, it is the process that minimizes errors with respect to changes in bandwidth.

Reviewer #2 (Remarks to the Author):

Dear authors

I sincerely submit my review report as an attached file.

The manuscript written by Sederberg et al. is the revised version which is transformed to Nature Communications from the previous submission to [REDACTED] [REDACTED]. Again, the development of direct sampling techniques of an optical field or a light waveform with petahertz bandwidth is one of the most important and broad-interest topics in not only the attosecond technology but also the broad optical science fields. The topic presented in this paper, therefore, matches the current scope of Nature Communications. Although several concepts for direct sampling techniques of an optical field, including attosecond streaking (ref. 26), petahertz oscilloscope (ref. 28), attosecond sampling of arbitrary optical waveforms (ref. 27), the tunneling ionization with a perturbation for the time-domain observation of an electric field (ref. 30), and the optical-field solid state detector (ref. 1), were demonstrated so far, their measurement results have not been evaluated by a comparison with another demonstrated method. Generally, it is significantly necessary to compare different methods for establishing new technology as reliable one. However, the comparison between different methods has been difficult task due to the experimental difficulty for attosecond temporal measurement setups.

In the previous revision, I pointed out the following three points as the new findings which Sederberg et al. concluded in this manuscript:

(1) The ability to retrieve the electric field waveform of light from a simple current measurement in a solid

state device.

(2) Its unprecedented bandwidth of detection.

(3) The ability to determine the temporal evolution of the charge carrier density from the measurement.

I made a reservation for [REDACTED] publication because I could not help but judged the item (1), which was the main argument in the previous manuscript, less impact compared with the first demonstration in Schiffrin et al. Nature 493, 70 (2013).

In the current revised version for Nature Communications, the authors sincerely revised my major concern.

We appreciate the reviewer's careful analysis of yet another iteration of this manuscript, and that he/she has concluded favorably regarding the revisions that we have made.

Especially, they add a clear discussion comparing the new main finding in the current work with the previous related works of refs. 1, 36, 37, in the middle part of the “Optical field sampling in solids” section in page 4. In this discussion, they clearly describe that they test and verify the relationship between the measured signal of the induced current and the driving field by comparing results of their NPS measurements to those of attosecond streaking and electro-optical sampling, which were well established methods of direct measurements of optical field oscillations. Although each technique, which corresponds to NPS, attosecond streaking, and electro-optical sampling, is previously demonstrated in ref. 1, ref. 38, and ref. 23, respectively, the combination of all the techniques is the first and valuable experimental demonstration for strengthening the functionality of the NPS method. In addition, the authors applied the NPS technique to five waveforms with frequencies ranging from <60 THz (5300 nm) to 1.1 PHz (275 nm) for demonstrating its unprecedented bandwidth of optical field sampling measurement. Such measurement, investigating the detection bandwidth of proposed method, is also the first demonstration in the direct optical-field sampling techniques reported previously. From this point of view, the manuscript is an important extension work of refs. 1, 36, and 37, which were reported by the authors’ related group of MPQ, in the attosecond science field community, while the technological concept of the detection scheme presented in the current manuscript cannot be said totally new.

In the following, minor comments are described.

- (1) The authors clearly describes the current signal, $S_d(\tau)$, as the convolution between the vector potential $A_d(t)$ and the gating function $G_d(t-\tau)$ in the current revised version. And also they comment that the noise level being higher for shorter wavelengths can be explained by the fact that, for a given amplitude of the electric field, the amplitude of $A_d(t)$ is proportional to the wavelength of the driving field. Although this description is reasonable, the signal amplitude $S_d(\tau)$ does not show the wavelength dependence at fixed power in Fig. 5(b). The author should comment on that shortly.

The reviewer is correct that the dependence of the signal on A_d suggests an amplitude proportional to the wavelength, and that this is not clearly reflected in the measurement vs. peak power. The reason for this is that the proportionality between A_d and its power (the main independently measurable quantity) is non-trivial: it depends on the focused spot size of the pulse, as well as the presence of any spatiotemporal distortions (which would not be surprising in the case of few-cycle pulses generated in a nonlinear crystal). On the extremes of the spectrum measured (the mid infrared and UV), the spot size measurements are rather difficult, as the sensitivity of CCDs is rather low, and easily contaminated with light from the laser driving the nonlinear generation of the pulses. Thus, the measurements vs. power are intended to indicate the degree of linearity of the measurement when only the power is changed during the experiment; a reliable

independent, calibrated measurement of $A_d(\omega)$ is unfortunately not possible with the equipment available. We have modified the relevant sentence in the caption to clarify this point.

- (2) In addition to the above comment related to Fig. 5(b), the slopes of the signal as a function of driving laser power for several wavelengths seems almost same. Does the linearity of this curves originate from the simple Ohm's law in the high-frequency region? Under the assumption of simple Ohm's law, the same slope indicates the same resistance for the electronic transport at wide range of frequency from 60 THz to 1 PHz. Does this indicates the scattering process of electrons driven by optical field shows no frequency dependence? It seems that such finding should be pointed out and discussed by the authors.

Unfortunately the measurement does not contain sufficient information to define quantities involved in transport - as can be seen in Fig. S3, the momentum relaxation time makes very little difference in the shape of the measured waveform, influencing primarily its amplitude. Essentially, when the current is following $A(t)$, rather than $E(t)$ as one would expect from a simple ohmic response, the transport is more ballistic in nature. We would not exclude that measurements such as these in the future may be designed to provide more insight into these quantities, but at the present moment, our calculations suggest that their influence on the measurement is too weak to allow for detailed conclusions about transport to be drawn, beyond the idea that the fact that the delay-dependent signal reproduces the vector potential is broadly consistent with the conductivity predicted by the Drude-Sommerfeld model.

In summary, I find that their description in the current manuscript does contain the sufficient novelty to justify a publication in Nature Communications.

We are grateful that the reviewer concludes in favor of publication of the present manuscript.

Reviewer #4 (Remarks to the Author):

Dear authors,

I want to apologize for the delay to submit my review. Even though the nonlinear photoconductive sampling approach was published before by your group, I consider that the manuscript has sufficient novel results to deserve publication in Nature Communications. Here are the reasons to support my decision:

- First, I am very impressed by the measured delay between NPS and EOS as a function of the field strength of the photoexcitation pulses. This brings novel physics compared with previous published work.
- Second, the results of NPS from the UV to the mid-infrared are very impressive and denotes a technological breakthrough in optical science.

We appreciate the consideration and the reviewer's conclusion in favor of publication of our work.

Regarding the comparison of NPS and EOS measurements, and the time delay as a function of field strength, I was disappointed by the short discussion of the manuscript. I consider this as an important physics milestone and the authors should better explained their results and even speculated on the origin of the delay. From the TDDFT, the authors have found an intrinsic delay. What is the physical origin of this intrinsic delay? Second, the authors are discussing in the manuscript that the additional delay comes from cross-phase modulation. I am surprised that increasing the field strength reduces the delay. The authors should clarify this paragraph on page 7.

The reviewer has brought up a very insightful point, and we appreciate that they consider this measurement to be a milestone. Our intention was precisely this: to measure the fundamental latency in electronic transitions in a solid, providing us with a complete understanding of the duration and timing of electronic transitions with respect to the incident field. Due to the complicated nature of the strong-field transition process, it is beyond the scope of this manuscript to assign a single physical process to the observed delay. The reviewer may be familiar with the extensive history of publications investigating photoemission delays in atomic systems. We feel that the complexity of interpreting a delay in electronic transitions in solids is of a similar magnitude and would require extensive experimental effort, which will be performed in the future.

Although TDDFT is a powerful technique that confirms our experimental results, it unfortunately does not provide immediate intuitive insight into the physics responsible for its results, leaving the underlying physics of the delay it predicts an open question. We have, however, added further discussion of the point to the SI, relating it to the observation in the above threshold ionization of argon.

Physically, a delay of this type suggests a phase shift between the current driven in the system by the laser field and the harmonic conjugate of the field; this can for example be influenced by the relative frequency difference between the field and the resonances of the system and the coherent nature of the excited state population.

The reduction of delay at higher field strengths has a simpler origin, however, which can be replicated simply through Maxwell's equations: the build up of free carriers causes an increase in reflectivity after the maximum of the field. This suppresses the intensity of the field after the carrier injection, causing the apparent first moment of the pulse transmitted

into the material to shift to earlier times. We have modified the relevant text to clarify this point.

Regarding the NPS measurements from the UV to the mid-infrared (Figure 5), I am surprised that the waveforms are looking like continuous functions, as compared to the results presented in Figure 4. Also, since those measurements arise from pump-probe scans, I expect that several scans have been acquired. It will be nice to see the repeatability of the measurements and some error bars, to judge the validity of the technique.

Indeed, the measurements are the result of pump-probe scans that are repeated multiple times and averaged. Thus, it is fairly straightforward to apply error bars depicting the standard error of the mean, as we have now added to the figure.

REVIEWERS' COMMENTS:

Reviewer #1 (Remarks to the Author):

Please accept my apologies for the delayed response.

The authors have addressed the criticisms of the reviewers and in light of their comments and modifications, I can recommend publication in Nature Communications, with the following small modifications to the manuscript.

1) The formula for the RMS error is not properly normalised and therefore the resulting number is meaningless (and it is *not* a percentage error). The correct formula should be of the form:

$\text{delta_RMS} = \sqrt{\frac{\sum(E_1 - E_2)^2}{[\sum(E_1^2) * \sum(E_2^2)]}}$ (cite Christophe Dorrer and Ian A. Walmsley, "Accuracy criterion for ultrashort pulse characterization techniques: application to spectral phase interferometry for direct electric field reconstruction," J. Opt. Soc. Am. B 19, 1019-1029 (2002)) and use the correct formula.

2) The term time-dependent density functional theory is first used at the start of the section "sub-femtosecond carrier injection" on page 5, but the acronym (TDDFT) is not defined until its later use on page 6, and there are several uses of the acronym in between these two scenarios. The authors should define TDDFT at first use at the start of the section on page 5.

Reviewer #2 (Remarks to the Author):

Dear authors

I sincerely reviewed your revised manuscript and your reply to my previous comments. My decision is described in detail in the attached file.

Sincerely yours,

The manuscript written by Sederberg et al. is the revised version which is transformed to Nature Communications from the previous submission to [REDACTED]. In the previous review, I found that their description in the revised manuscript contained the sufficient novelty to justify a publication in Nature Communications, although I gave two minor comments for revision at that time.

For the first comment, I pointed out that the signal amplitude $S_d(\tau)$ does not show the wavelength dependence at fixed power in Fig. 5(b) although $S_d(\tau)$ should be proportional to the vector potential $A_d(t)$. The authors agreed this discrepancy and replied that the reason for this is that the proportionality between $A_d(t)$ and its power is non-trivial: it depends on the focused spot size of the pulse, as well as the presence of any spatiotemporal distortions, and that the measurements vs. power showed in Fig. 5(b) are intended to indicate the degree of linearity of the measurement when only the power is changed during the experiment. This authors' reply is reasonable because it is not generally easy to measure the power density, which corresponds to the vector potential in this experiment. Since the authors added the description explaining this situation in the caption of Fig. 5 in the re-revised version, my concern has been resolved by their revision.

For my second comment, I pointed out that the authors should comment the reason why the slopes of the signal vs. the driving laser power for several wavelengths seems almost same. The authors replied that the measurement does not contain sufficient information to define quantities involved in transport. And also they showed that the momentum relaxation time made very little difference in the shape of the measured waveform in Fig. S3 in their supplementary information. Although the experimental results showing the almost same slope for different wavelength optical waveform measurement is very interesting, I agreed that it is difficult to conclude perfectly the reason from the current experiment. From this point of view, the authors reply to my second comment is satisfied.

In summary, the authors' reply and revision does contain the sufficient novelty to justify a publication in Nature Communications.

Katsuya Oguri, Dr.
Group leader,
Quantum Optical Physics Research Group
Optical Science Laboratory
NTT Basic Research Laboratories, NTT Corporation

Reviewer #4 (Remarks to the Author):

Dear authors,

The manuscript has been carefully revised addressing all my comments and suggestions. I strongly recommend publication at Nature Communications

Sincerely yours

We thank the reviewers for their positive decision regarding our manuscript, and the time invested over the course of the extended review process. We have addressed the two final points raised by reviewer 1 through two small revisions to the manuscript, and respond specifically to the points below. The changes made in response to these comments and those of the editor are marked in the files `diffMaintext.pdf` and `diffSI.pdf`.

Reviewer #1 (Remarks to the Author):

Please accept my apologies for the delayed response.

The authors have addressed the criticisms of the reviewers and in light of their comments and modifications, I can recommend publication in Nature Communications, with the following small modifications to the manuscript.

1) The formula for the RMS error is not properly normalised and therefore the resulting number is meaningless (and it is *not* a percentage error). The correct formula should be of the form:

$\text{delta_RMS} = \sqrt{\frac{\sum(E_1 - E_2)^2}{\sum(E_1^2) * \sum(E_2^2)}}$ (cite Christophe Dorrer and Ian A. Walmsley, "Accuracy criterion for ultrashort pulse characterization techniques: application to spectral phase interferometry for direct electric field reconstruction," J. Opt. Soc. Am. B 19, 1019-1029 (2002)) and use the correct formula.

We have used the calculation from the reference suggested by the reviewer and cited that publication. We have carried out the normalization used by Dorrer and Walmsley, such that $\|E\|=1$, and the final deviation is a unitless value. The formula from the reviewer contains an error such that it has units of m/V (the fields should be normalized individually,

$$\varepsilon = \left[\sum \left(\frac{E_1}{\sqrt{\sum E_1^2}} - \frac{E_2}{\sqrt{\sum E_2^2}} \right)^2 \right]^{1/2},$$

which is the formulation in the reference with the normalization made explicit, and results in a unitless ε).

2) The term time-dependent density functional theory is first used at the start of the section "sub-femtosecond carrier injection" on page 5, but the acronym (TDDFT) is not defined until its later use on page 6, and there are several uses of the acronym in between these two scenarios. The authors should define TDDFT at first use at the start of the section on page 5.

We now define TDDFT in the place where it first appears in the text.